# Metheor: Ultrafast DNA methylation heterogeneity calculation from bisulfite read alignments

Dohoon Lee[1,2], Bonil Koo[3], Jeewon Yang[4], Sun Kim[3,4,5,6¤] *

**1** Bioinformatics Institute, Seoul National University, Seoul, Republic of Korea, **2** BK21 FOUR Intelligence Computing, Seoul National University, Seoul, Republic of Korea, **3** Interdisciplinary Program in Bioinformatics, Seoul National University, Seoul, Republic of Korea, **4** Interdisciplinary Program in Artificial Intelligence, Seoul National University, Seoul, Republic of Korea, **5** Department of Computer Science and Engineering, Seoul National University, Seoul, Republic of Korea, **6** MOGAM Institute for Biomedical Research, Yong-in, Republic of Korea

¤ Current address: Department of Computer Science and Engineering, Seoul National University, Seoul, Republic of Korea
* sunkim.bioinfo@snu.ac.kr

**Data Availability Statement:** Metheor is freely available for any users under the GPL-3.0 license. The source code can be anonymously downloaded at the GitHub repository (https://github.com/dohlee/metheor), and the executable is distributed through conda package manager to facilitate the

## Abstract

Phased DNA methylation states within bisulfite sequencing reads are valuable source of information that can be used to estimate epigenetic diversity across cells as well as epigenomic instability in individual cells. Various measures capturing the heterogeneity of DNA methylation states have been proposed for a decade. However, in routine analyses on DNA methylation, this heterogeneity is often ignored by computing average methylation levels at CpG sites, even though such information exists in bisulfite sequencing data in the form of phased methylation states, or methylation patterns. In this study, to facilitate the application of the DNA methylation heterogeneity measures in downstream epigenomic analyses, we present a Rust-based, extremely fast and lightweight bioinformatics toolkit called Metheor. As the analysis of DNA methylation heterogeneity requires the examination of pairs or groups of CpGs throughout the genome, existing softwares suffer from high computational burden, which almost make a large-scale DNA methylation heterogeneity studies intractable for researchers with limited resources. In this study, we benchmark the performance of Metheor against existing code implementations for DNA methylation heterogeneity measures in three different scenarios of simulated bisulfite sequencing datasets. Metheor was shown to dramatically reduce the execution time up to 300-fold and memory footprint up to 60-fold, while producing identical results with the original implementation, thereby facilitating a large-scale study of DNA methylation heterogeneity profiles. To demonstrate the utility of the low computational burden of Metheor, we show that the methylation heterogeneity profiles of 928 cancer cell lines can be computed with standard computing resources. With those profiles, we reveal the association between DNA methylation heterogeneity and various omics features. Source code for Metheor is at https://github.com/dohlee/metheor and is freely available under the GPL-3.0 license.

public use of the software (https://anaconda.org/dohlee/metheor). We also provide the codes for the simulation of WGBS reads at the dedicated GitHub repository (https://github.com/jwyang21/simulate_WGBS). Finally, to facilitate such future studies on the DNA methylation heterogeneity in cancers, we made the DNA methylation heterogeneity profiles of 928 CCLE cell lines computed by Metheor publicly available through Figshare (https://doi.org/10.6084/m9.figshare.21100717.v1).

**Funding:** This research was supported by the Bio & Medical Technology Development Program of the National Research Foundation (NRF) funded by the Ministry of Science & ICT(NRF-2019M3E5D307337511 and NRF-2022M3E5F3085677) (to S.K.) and Institute of Information & communications Technology Planning & Evaluation (IITP) grant funded by the Korea government(MSIT) [NO.2021-0-01343, Artificial Intelligence Graduate School Program (Seoul National University)] (to S.K.). The funders had no role in study design, data collection and analysis, decision to publish, or preparation of the manuscript.

**Competing interests:** The authors have declared that no competing interests exist.

## Author summary

DNA methylation is the most extensively studied epigenetic modifications that plays a pivotal role in key biological processes. The advance of next-generation sequencing technology combined with bisulfite treatment of DNA, namely bisulfite sequencing, allowed fine-grained characterization of DNA methylation states at basepair-resolution, and it facilitated researchers elucidate the significance of DNA methylation at various genomic contexts. To date, most of the routine DNA methylation analyses only deal with per-CpG methylation levels. That is, they compute the proportion of CpGs in methylated state for each CpG throughout the genome. However, bisulfite sequencing data harbor information beyond individual CpG methylation states: methylation states co-occurring in a single sequencing read. This information is important because such 'phased' methylation states can inform us about the epigenetic diversity of cell populations as well as the local regulation states of the epigenome. We collectively refer to those information as DNA methylation heterogeneity. In this study, we present a Rust-based software named Metheor for ultrafast and accurate calculation of DNA methylation heterogeneity from bisulfite sequencing data. We show that Metheor reduces execution time and memory footprint up to 300-fold and 60-fold, respectively, compared to existing implementations, and analyze DNA methylation heterogeneity profiles of 928 cancer cell lines.

## Introduction

Unlike genomes, changes in epigenomes are dynamic and reversible. This plasticity has been increasingly highlighted over a recent decade, since the spatiotemporal dynamics of epigenetic modification are shown to form the basis of diverse cellular processes such as cell differentiation and senescence. The variability of epigenomes is the key mechanism that cells acquire diverse and precise functions throughout the whole body of an individual to sustain the life of an organism. Furthermore, the dysregulation of the epigenetic variability has shown great biological implication for various cancer types, as it increases the adaptive potential of a cancer cell population against treatments [1].

Among the diverse manifestations of epigenetic variabilities, cell-to-cell variability of DNA methylation states is one of the most actively investigated research topics. To measure the diversity of bulk cell population in terms of DNA methylation, it is necessary to identify the epigenetic configurations of individual cells. Single-cell bisulfite sequencing can directly resolve this challenge, but its cost makes it hardly applicable to a large-scale study. An effective alternative is to extract DNA methylation states co-occurring in a single sequencing read, and consider each pattern as a pseudo-barcode that identifies each cell. By measuring the diversity of DNA methylation patterns aligned at each genomic region, we can obtain a partial estimate of the true epigenetic diversity across cell population.

Despite many proof-of-concept experiments underscoring the utility of those DNA methylation heterogeneity measures in physiopathological conditions, a highly efficient toolkit for quantifying the extent of the heterogeneity is still lacking. To facilitate a large-scale functional study of DNA methylation heterogeneity based on bisulfite sequencing data, we developed a fast and lightweight software called *Metheor*. In this study, we present the functionality of Metheor and benchmark its performance against existing code implementations for DNA methylation heterogeneity measures. Also, to demonstrate the utility of Metheor in large-scale studies, we provide DNA methylation heterogeneity analyses of 928 cancer cell lines from

Cancer Cell Line Encyclopedia (CCLE), which reveal novel association between the extent of epigenetic heterogeneity and other omics features. We believe this software will serve as a convenient toolkit helping researchers fully utilize the phasing information of DNA methylation states that have not been investigated actively so far.

## Design and implementation

### Functionality and implementation

Given a bisulfite read alignment file, Metheor can efficiently compute all of the six DNA methylation heterogeneity measures proposed to date [2–6] (Fig 1A and 1B and S1–S4 Figs). In short, epipolymorphism and methylation entropy quantify the diversity of DNA methylation patterns, or epialleles, in a given cell population. Fraction of discordant read pairs (FDRP) and quantitative FDRP (qFDRP) measure the epiallelic diversity at CpG resolution. On the other hand, proportion of discordant reads (PDR) measures the extent of local homogeneity of DNA methylation through computing the proportion of bisulfite sequencing reads having two different DNA methylation states at the same time. Methylation haplotype load (MHL) similarly captures the local homogeneity of DNA methylation status by quantifying how well a methylation haplotype (i.e., a stretch of consecutively methylated CpGs) is conserved throughout the cell population. For the detailed description for each of the measures and the algorithms to compute them, please refer to the S1 Text.

The main algorithmic advantage of Metheor is that it is a "one-sweep" algorithm that iterates through the entire sequencing reads only once (Fig 1C). On the other hand, a benchmark counterpart R package, WSHPackage [6], iterates through a set of CpGs specified by a user and fetches the reads covering each CpG using indexed alignment file (Fig 1D). As the most time-consuming operation in this setting is fetching sequencing reads from alignments, we can estimate the time complexity of the methods using the total number of sequencing read accesses. For Metheor, the estimation is trivial; it requires $n$ sequencing read accesses in total, where $n$ is the number of aligned reads. For WSHPackage, the number of sequencing read accesses is $\lambda \times n$, where $\lambda$ is the average number of CpGs within a single sequencing read. Therefore, the performance advantage of Metheor compared to WSH will be determined by the coefficient $\lambda$. Importantly, the empirical distribution of CpGs throughout the genome is uneven and known to form CpG-dense regions including CpG-islands. Reduced representation bisulfite sequencing (RRBS) predominantly targets those CpG-dense regions, so the coefficient $\lambda$ for RRBS experiment is generally greater than 1 (Fig 1E). This is why the read-centric algorithm used in Metheor empirically runs faster than CpG-centric ones.

Note that when a user wants to compute DNA methylation heterogeneity levels for only a subset of CpGs rather than the whole set of CpGs, it may be disadvantageous for Metheor since it iterates all the aligned reads while CpG-centric approaches uses alignment index to access only to CpGs specified by users. However, we observed that Metheor still ran faster than WSHPackage for PDR, FDRP and qFDRP calculation even when only a subset of CpGs are considered ($3 \sim 4$-fold speedups when 5% of CpGs are considered; S5 Fig), underscoring the efficiency of the implementation of Metheor.

Metheor also supports alignments generated not only by Bismark, but also other widely used methylation-aware aligners, by providing a subcommand to attach a tag denoting the methylation states of cytosines to each aligned read (Fig 1A). This warrants the wide applicability of Metheor at the downstream of various bisulfite read processing pipelines. Metheor is implemented in Rust language and distributed via the conda package manager.

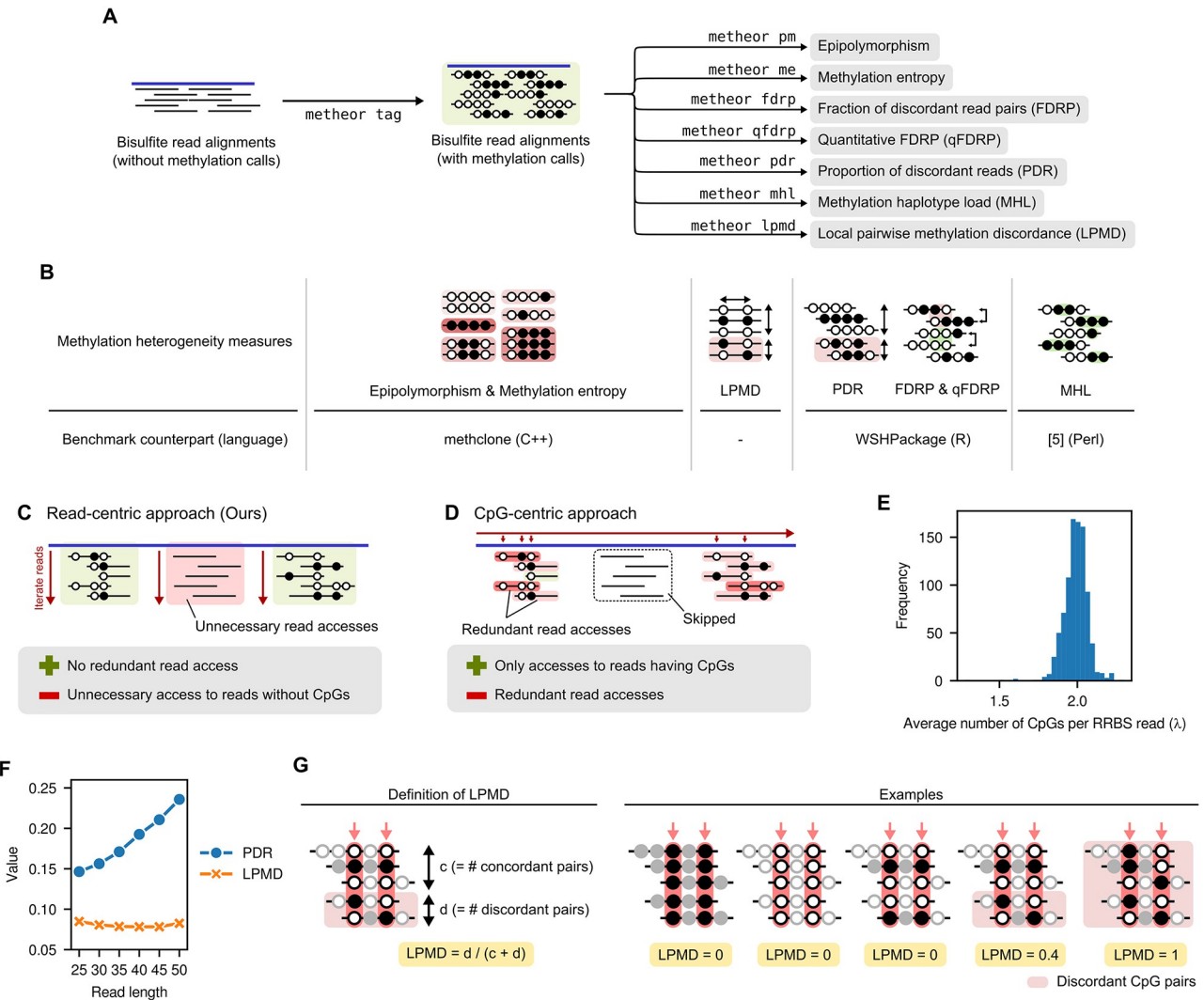

**Fig 1. Overview of Metheor.** (A) The input for Metheor is bisulfite read alignment tagged with Bismark methylation call strings. Using each of the seven subcommands shown, Metheor computes the corresponding DNA methylation heterogeneity measure. If reads were aligned with a tool other than Bismark, Metheor can still add tag for methylation call string with `metheor tag` subcommand to make alignment file compatible for Metheor run. (B) Schematic diagram for DNA methylation heterogeneity measures and benchmark settings in this study. [5] denote the Perl script provided by the authors along with the article proposing the utility of MHL. (C, D) Schematic diagram illustrating (C) read-centric algorithm and (D) CpG-centric algorithm for the computation of DNA methylation heterogeneity. The advantages (plus symbol) and disadvantages (minus symbol) are shown below the diagrams. (E) Distribution of the average number of CpGs per sequencing read for the RRBS data from 928 CCLE cell lines. (F) Genomewide average levels of proportion of discordant reads (PDR) and local pairwise methylation discordance (LPMD) against varying read lengths. (G) Schematic illustration for the definition of local pairwise methylation discordance (LPMD) and examples. The proportion of reads having different DNA methylation states for a pair of CpGs (red arrows) are computed.

## Definition of local pairwise methylation discordance (LPMD)

To ameliorate the read length bias of PDR, we propose a new measure called local pairwise methylation discordance (LPMD) in this study (Fig 1G). LPMD considers the discrepancy of methylation states between a pair of CpGs placed at a fixed distance from each other instead of the read-wise classification of DNA methylation discordance in PDR. As in the computation of PDR, the concordance of DNA methylation states are only examined for pairs of CpG methylation states that are present on the same sequencing read. Therefore, LPMD allows a

consistent comparison between any sequencing experiments regardless of read length, as long as the same range of CpG pair distance is considered. Throughout this study, we computed the LPMD values using CpG pairs that are 2bp to 16bp away from each other regarding the length of CCLE RRBS reads (29nt). We note that the trend of LPMD values are robust to the choice of the genomic distance window between CpG pairs (S6 Fig), so in general $2 \sim 16$bp window is a reasonable choice.

## Simulation of sequencing data with different read lengths

In the computation of PDR, a read is classified as discordant if it contains at least one CpG pair with different methylation states. Therefore, it is conceivable that the levels of PDR will increase as the length of sequencing read gets longer because the mere chance of observing a discordant CpG pair increases as more CpGs are considered. This is not desirable since it hampers the comparison of the metric between the sequencing experiments of different read lengths, which is often unavoidable for large-scale collaborative studies or meta-analyses. To examine the read length bias of PDR and LPMD, we simulated sequencing data with different read lengths by manipulating publicly available RRBS data from Ewing sarcoma tumor sample. The raw 50bp-sequencing reads were downloaded under SRA run accession SRR5222549, and aligned to hg38 reference genome using Bismark v0.23.1. For each aligned read, the methylation call string generated by Bismark was trimmed from 3'-end to mimic reads of shorter length. Read alignment files with 25bp, 30bp, 35bp, 45bp and 50bp reads were separately generated and genomewide average PDR values were computed. As expected, we observed the clear read length bias for PDR, scaling linearly to increasing read lengths (Fig 1F).

## Performance benchmark

We compared the performance of our method with existing methods (methclone [7], WSHPackage [6] and perl scripts from [5]) for the six existing measures. Three different types of simulated sequencing datasets with varying numbers of reads were used (S1 Text). In brief, we generated two RRBS datasets (by simulation and real-world data subsampling) and pseudo-whole genome bisulfite sequencing experiments reproducing the methylation erosion scenario as proposed in [6]. All the benchmark experiments were done on a server with Intel (R) Xeon(R) E7–4850 2.10 GHz CPU and 512GB of RAM.

## Demonstration of the validity of results

To verify that Metheor produces methylation heterogeneity levels accurately, we compared CpG-wise and CpG quartet-wise methylation heterogeneity levels produced by Metheor to those produced by reference implementations. For experiment, we used the results from simulated RRBS data with 20M reads. Note that PDR, MHL, FDRP and qFDRP levels were computed for each individual CpG, and PM and ME were computed for each CpG quartet.

## Computation of DNA methylation heterogeneity profiles of CCLE cell lines

We downloaded raw RRBS sequencing reads for 928 CCLE cell lines under SRA study accession SRP186687. RRBS reads were preprocessed using Trim Galore! v0.6.7 with $--$rrbs option and aligned to hg38 reference genome using Bismark v0.23.1. Given the read alignments, genomewide and region-specific DNA methylation heterogeneity profiles were obtained using Metheor v0.1.2 with default parameters. Of note, genomic contexts of interest included CpG islands, CpG shores, CpG shelves, DNA methylation canyons, long and short interspersed nuclear elements (LINEs and SINEs, respectively), long terminal repeats (LTRs),

exons, introns, gene bodies and promoters of protein coding genes. Annotations for CpG islands and transposable elements are downloaded from UCSC table browser. CpG shores were defined as 2kbp regions flanking CpG islands on their both side, but regions overlapping with any CpG islands were excluded. Similarly, CpG shelves were defined as 2kbp regions flanking the stretches of CpG islands and shores. GENCODE v38 gene annotations were used to define exons, introns, gene bodies and promoters, where promoters were defined as 2kbp regions centered at TSS for each protein coding gene. Annotations for DNA methylation canyons were downloaded from the supplementary material of [8].

## Results

### Execution time and memory usage benchmarks

We observed significant speedups (up to 300-fold depending on measure and data size) for the computation of all DNA methylation heterogeneity measures (Fig 2A and 2B and S7 Fig). It is worth noting that the total running time of Metheor and WSHPackage were both linearly proportional to the number of sequencing reads in the input data. Especially, we could achieve extreme speedup for FDRP and qFDRP calculation using reservoir sampling-based approach (S1 Text). Since the utility of FDRP and qFDRP has been limited by the slow computation [6], we expect that Metheor will facilitate the use of those measures in further experimental setups. We also observed significant reduction in memory footprints (Fig 2C and 2D and S7 Fig).

WSHPackage allows PDR, FDRP and qFDRP to be calculated using multiple threads. To examine whether it can achieve performance comparable to Metheor when multiple threads are used, we particularly focused on the three measures, PDR, FDRP and qFDRP, for which WSHPackage allows the computation to be multithreaded. The running time for the computation of the three measures was measured using 20M simulated RRBS reads, with three replicates using each of 4, 8, 16 and 32 threads for WSHPackage. Surprisingly, we found that Metheor, only with a single thread, showed remarkable performance improvement ($13.7 \sim 175.8$-fold faster) even compared to WSHPackage using 32 threads (Table 1).

### Validity of the result

We verified that the DNA methylation heterogeneity levels computed by Metheor are consistent with the results from the reference implementations at both individual CpG or epiallele level (Fig 2E). Specifically, we observed exactly identical results for PDR, MHL, PM and ME levels. For FDRP and qFDRP, values from Metheor and WSHPackage were not exactly the same because these measures depend on random sampling of sequencing reads and it is not possible to reproduce the random sampling process of the reference implementation. Nevertheless, they showed extremely high and significant correlation with each other, supporting that Metheor successfully implements procedures to compute FDRP and qFDRP in desired ways. Altogether, these results ensure the reliability of the results from Metheor throughout all the measures it supports. Of note, Metheor uses 0-based half-open inteval system that comply with standard BED format. All the comparisons for the validation were done after translating the coordinates of the outputs appropriately.

### Exploratory analysis on the DNA methylation heterogeneity profiles of 928 CCLE cell lines

**Metheor allows the large-scale characterization of DNA methylation heterogeneity profiles in standard computing resources.** To demonstrate the utility of Metheor for a large-scale DNA methylation heterogeneity studies, we applied Metheor to compute the genomewide

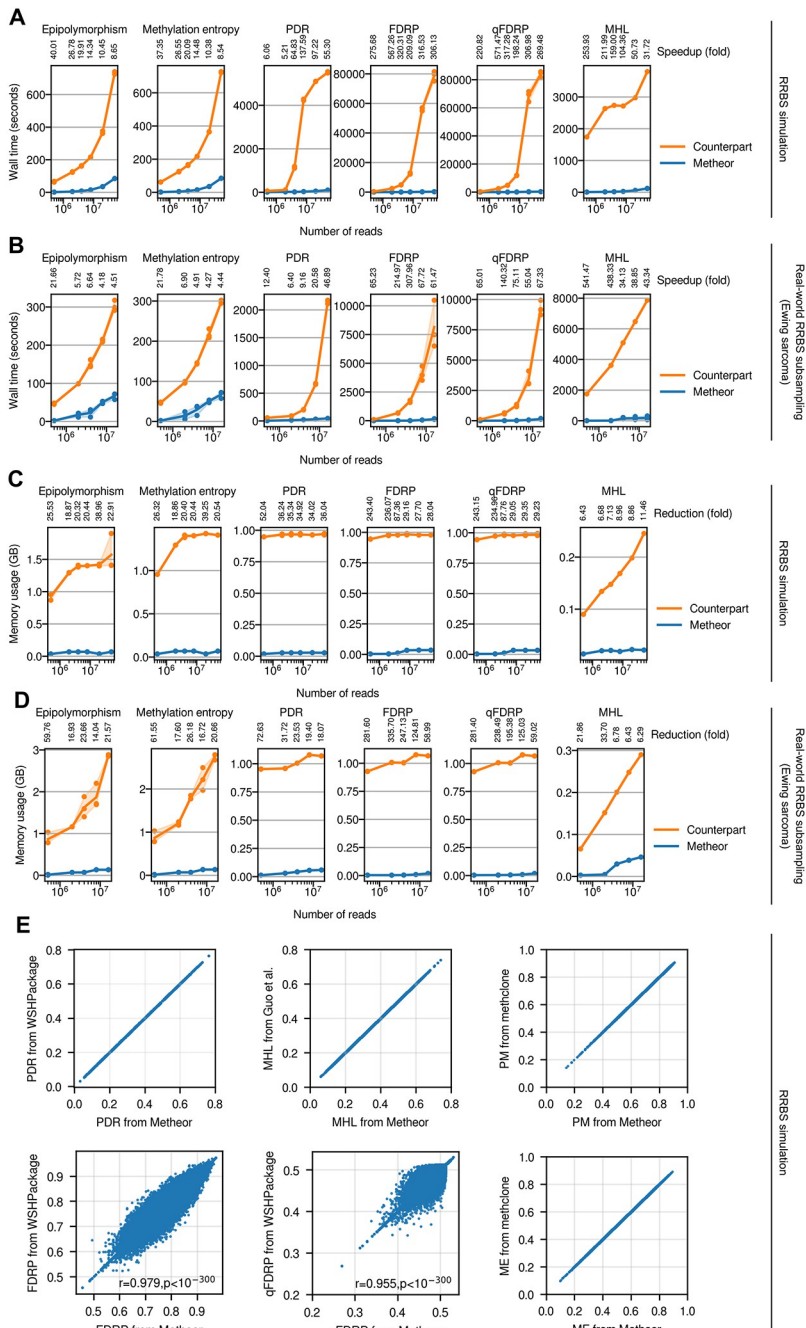

**Fig 2. Performance benchmark and validity of the results.** Benchmarking the running time of Metheor using (A) simulated RRBS dataset and (B) Ewing sarcoma RRBS dataset. Values below the name of each of the measures denote the amount of speedup (in fold) in Metheor compared to its benchmark counterpart. Benchmarking the memory usage of Metheor using (C) simulated RRBS dataset and (D) Ewing sarcoma RRBS dataset. Values below the name of each of the measures denote the amount of memory usage reduction (in fold) in Metheor compared to its benchmark counterpart. All the benchmark experiments were repeated for three times, except for MHL. Lines denote the average wall time and shades represent the 95% confidence interval. The wall time for MHL computation was measured for only once. (E) Validity of the results. CpG-wise (PDR, MHL, FDRP and qFDRP) and CpG quartet-wise (PM and ME) methylation heterogeneity levels were compared between Metheor and the corresponding reference implementations. Pearson's correlation coefficient and corresponding p-values are shown for FDRP and qFDRP.

**Table 1. Performance comparison with WSHPackage using multiple threads for 20M RRBS-simulated reads.**

| Tool | # threads | PDR (fold) | FDRP (fold) | qFDRP (fold) |
|---|---|---|---|---|
| **Metheor** | **1** | **0.87 (1)** | **2.96 (1)** | **3.72 (1)** |
| WSHPackage | 4 | 34.03 (38.92) | 1813.86 (612.65) | 1832.19 (492.70) |
| | 8 | 19.57 (22.38) | 1072.79 (362.35) | 1097.32 (295.09) |
| | 16 | 11.21 (12.82) | 652.74 (220.47) | 671.59 (180.60) |
| | 32 | 11.94 (13.66) | 520.58 (175.83) | 527.40 (141.83) |

Values denote the wall time in minutes to compute each measure. Fold speedups in corresponding run of Metheor are shown in parentheses.

and context-specific DNA methylation heterogeneity profiles (PDR, ME, PM, FDRP, qFDRP, MHL and LPMD) using RRBS data for 928 cancer cell lines from CCLE. Of note, we restricted our processing pipeline to use at most 32 cores, each of which being assigned to single-threaded Metheor run, to show that Metheor can be run in standard computing resources that are available to a majority of the researchers, especially including experimental biologists.

The computation of PDR, ME, PM and MHL for each cell line took 99.75, 144.52, 97.54 and 182.15 seconds on average, respectively, and the whole pipeline for 928 cell lines did not take more than one and an half hour. Since the running time estimation of four-threaded WSHPackage run for PDR yields about 5 days according to Table 1, it shows the significant reduction of the computational burden of Metheor. Of note, the computation of LPMD took 397.55 seconds on average for each cell line. Furthermore, the computation of FDRP and qFDRP took 410.36 and 410.19 seconds on average, respectively, making the whole pipeline finish in about 3.3 hours. As the four-threaded WSHPackage runs for FDRP and qFDRP are expected to take 338 and 271 days in this setting, respectively, the use of high-performance computing cluster would have been almost necessary to compute these measures. Altogether, these results directly show the wide applicability of Metheor for large-scale DNA methylation heterogeneity studies.

In the following sections, we provide some exploratory analyses on the biological significance of the DNA methylation heterogeneity profiles of 928 cancer cell lines characterized by Metheor.

**Characteristics of the local DNA methylation disorder captured by LPMD across 928 cancer cell lines.** We first analyzed the genomewide average levels of LPMD across the 928 cancer cell lines according to their tissues of origin (Fig 3A) and disease types (Fig 3B). Overall, we could not observe consistent trends shared between genomewide methylation levels (i.e., average methylation level across all CpGs throughout the genome) and genomewide LPMD levels across tissues and diseases. This implies that the observed DNA methylation heterogeneity levels are not a mere consequence of stochastic DNA methylation, but at least there exist some tissue-specific regulatory mechanism that constrains the local homogeneity of DNA methylation states.

The genomewide methylation and LPMD levels were highly variable within the group of cell lines derived from haematopoietic and lymphoid tissues (Fig 3A), reflecting the heterogeneous disease composition (Fig 3B). Notably, malignancies derived from B cells showed highest genomewide LPMD, but they did not show consistent levels of global DNA methylation. The increased local disorder of DNA methylation states in B cell-derived malignancies is emphasized when the acute lymphoblastic B cell leukaemia and acute lymphoblastic T cell leukaemia are compared. This is particularly interesting since the importance of DNA methylation heterogeneity has been already highlighted in multiple myelomas [9] and diffuse large B cell lymphoma [10, 11].

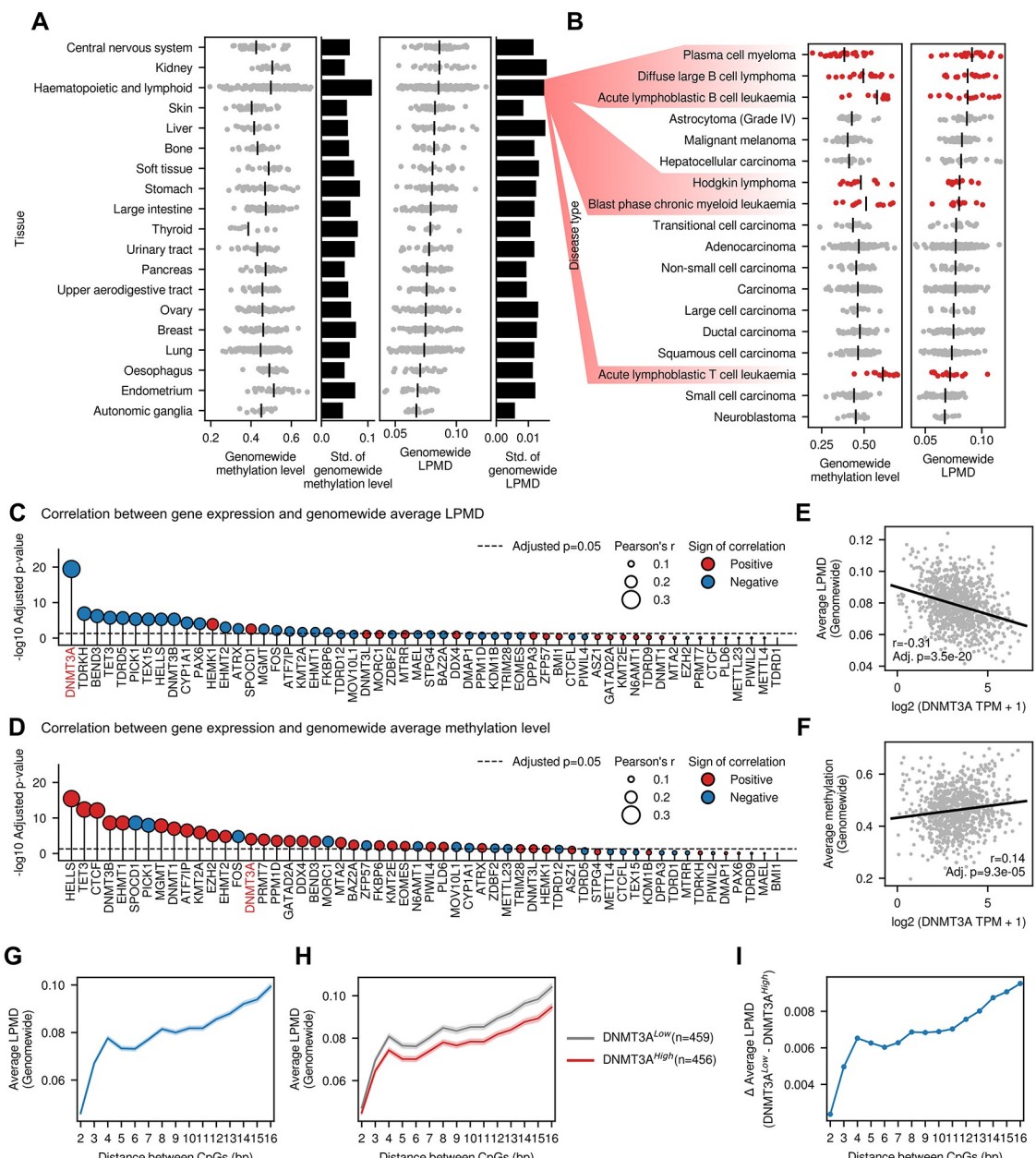

**Fig 3. Characteristics of LPMD across 928 cancer cell lines.** (A) Genomewide average methylation levels and LPMD levels grouped by tissue types. Black vertical lines denote groupwise average levels of methylation and LPMD levels. Black horizontal bars on the right side denote the standard deviation of corresponding values. (B) Genomewide average methylation levels and LPMD levels grouped by disease types. Disease types from haematopoietic and lymphoid tissues are highlighted in red. (C, D) Correlation between mRNA expression and (C) genomewide average LPMD or (D) genomewide average methylation level. Genes are ranked according to the p-values of the corresponding correlation coefficients. P-values were adjusted using Benjamini-Hochberg procedure. (E, F) Correlation between DNMT3A expression and (E) genomewide average LPMD or (F) genomewide average methylation level. (G, H) Trends of fixed-distance average LPMD values. Shades denote 95% confidence interval. In (H), Cell lines were divided into two groups based on the median DNMT3A expression. (I) Difference of fixed-distance average LPMD values between DNMT3A$^{High}$ and DNMT3A$^{Low}$ groups.

To reveal potential epigenetic regulators of the genomewide level of the local disorder of DNA methylation, we narrowed down our focus to a set of genes associated with GO term 'DNA methylation' (GO:0006306). Among the 59 genes associated with the term, we examined 56 genes whose mRNA expression levels were available. For each gene, the correlation between its mRNA expression level and genomewide LPMD or DNA methylation level were computed across the cancer cell lines, and we observed moderate negative correlation between *DNMT3A* expression and LPMD levels (Fig 3C and 3D). Interestingly, the correlation between DNMT3A expression and global DNA methylation level was not prominently stronger than the other genes (Fig 3E and 3F). As a measure of the local disorder of DNA methylation, an advantage of LPMD compared to PDR is that it explicitly takes the distance between a pair of CpGs into account. We asked whether the probability of observing discordance of DNA methylation states increases with increasing distance between a pair of CpGs (Fig 3G). In other words, we examined the local homogeneity of individual DNA methylation states in a quantitative manner. As a result, we verified that the discordance of DNA methylation states were positively correlated with the distance between CpGs. We confirmed that there were sufficient number of CpG pairs for the computation of LPMD (S8 Fig). Intriguingly, when the cell lines were divided into two groups based on their *DNMT3A* mRNA expression levels, we observed significant differences of LPMD levels between the two groups (Fig 3H, Two-tailed independent t-test $p < 0.001$ for all individual distances), and the absolute difference increased along the distance (Fig 3I). We leave the elucidation of the exact biological mechanism underlying the association between the *DNMT3A* expression and the local disorder of DNA methylation as a future work, nevertheless we hypothesize that the processive *de novo* DNA methylation by DNMT3A dependent on the oligomeric state of the protein [12] may play a critical role in this phenomenon. Specifically, homotetrameric DNMT3A shows processive *de novo* methylation, where the methylation is added to consecutive CpGs. On the other hand, homodimeric DNMT3A shows distributive catalysis, where the enzyme complex frequently dissociates from the DNA. As the oligomeric state of homomer complex is known to heavily dependent on monomer concentration [13], we speculate that the increase of DNMT3A gene expression will make intracellular DNMT3A oligomeric state biased toward homotetramers, thus increasing the local homogeneity of DNA methylation states.

**Homogeneity of DNA methylation patterns across cell population is associated with the stemness of cancer cells.**   Having analyzed the intracellular DNA methylation disorders using LPMD, we then tried to systematically identify genes whose expression levels are associated with the global extent of intercellular DNA methylation heterogeneity using methylation entropy. To this end, for every gene, we computed the correlation between its expression and the average methylation entropy levels at the promoters of all protein coding genes (Fig 4A). Interestingly, we observed that the genes whose expressions were negatively correlated with global promoter methylation entropy exhibited considerable enrichment for biological processes including extracellular matrix organization, cell migration, cell differentiation and epithelial to mesenchymal transition (Fig 4B) and it was true for the other DNA methylation heterogeneity measures including LPMD, PDR, epipolymorphism, FDRP and MHL (S9 Fig (A)). These terms collectively imply that the homogeneity of DNA methylation patterns is associated with the increased metastatic potential as well as increased stemness of cancer cells. As expected, cancer cell lines derived from metastatic origins displayed lower promoter methylation entropy levels (Fig 4C, $p = 0.0478$) and other DNA methylation heterogeneity levels (S9 Fig (B)). Fig 4D shows weak, but significant negative correlation for the two representative genes within canonical Wnt signaling (*WNT7A* and *CTNND2*). Furthermore, the activity of Wnt signaling pathway, inferred by bioinformatic method called subsystem activation score [14], also showed negative correlation with global levels of promoter methylation entropy (Fig

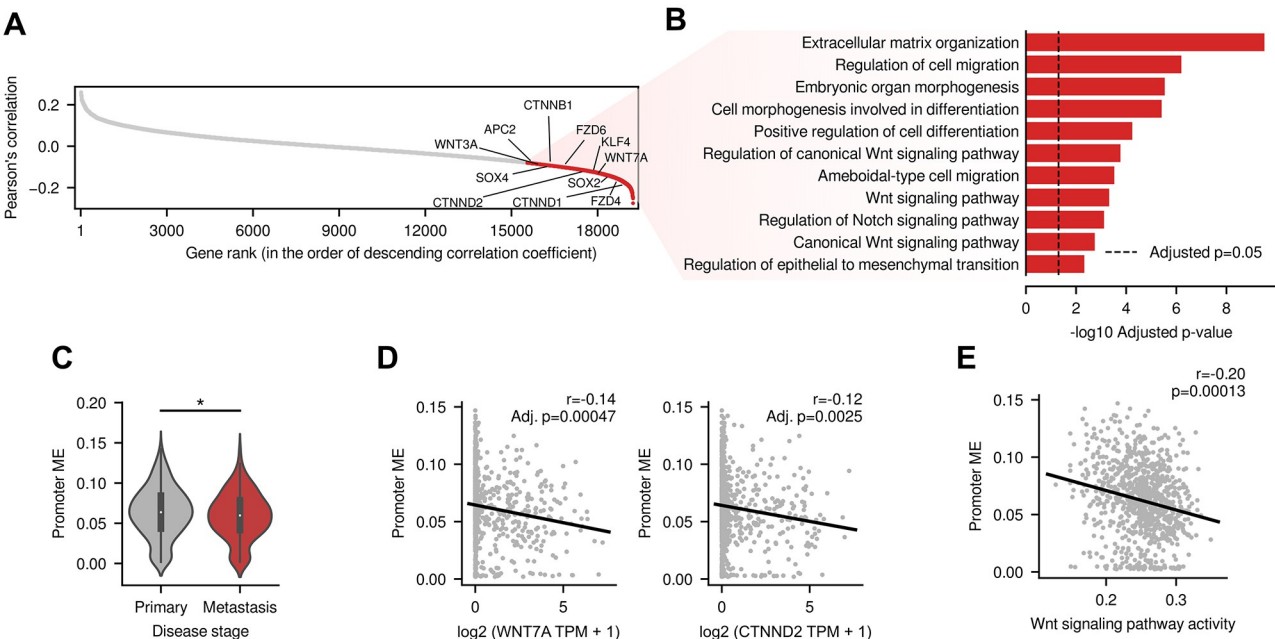

**Fig 4. Association between methylation entropy and cancer stemness.** (A) Genes were ranked by the Pearson's correlation between their expression and average methylation entropy levels across promoters. Red dots represent 3,680 genes having statistically significant correlations (Benjamini-Hochberg adjusted p-value < 0.05), and the results of functional enrichment analysis using those genes are shown in (B). (C) The distribution of promoter methylation entropy levels in primary and metastatic cancer cell lines. (D) The association between promoter methylation entropy levels and two genes representative of Wnt signaling pathway (*WNT7A* and *CTNND2*). (E) The association between promoter methylation entropy levels and the activity of Wnt signaling pathway. *two-tailed independent t-test p < 0.05; In D-E, Pearson's correlation coefficients and associated p-values are shown. In D, p-values were adjusted using Benjamini-Hochberg procedure.

4E) and other DNA methylation heterogeneity levels (S9 Fig (C)). These results are noticeable in that stem cells in normal physiological condition show remarkable homogeneity of DNA methylation patterns [4]. Although the causal relationship between the observed DNA methylation heterogeneity in a cancer cell population and stem-like reprogramming remains to be studied, our results suggest the potential of the cancer cell population-level DNA methylation homogeneity as a new prognostic marker.

## Availability and future directions

The DNA methylation heterogeneity profiles generated by Metheor can be utilized to discover novel biomarkers when combined with functional genomic analyses. For example, the heterogeneity, or diversity, of DNA methylation patterns within a cancer cell population can serve as a predictive biomarker for certain anticancer drugs. Especially, through a preliminary analysis we found that PDRs at the promoters of tumor suppressor genes (TSGs) are significantly lower than that of oncogenes and the rest of the genes (S10 Fig), suggesting the different role of epigenetic regulation of TSGs and oncogenes in the context of cancer evolution. Moreover, the local heterogeneity of DNA methylation states may sensitize cancer cells to the perturbation of certain genes. That is, the configuration of local DNA methylation states may reflect the genetic dependency of cancer cells. Meanwhile, our exploratory analysis of DNA methylation heterogeneity reveals a link between the extent of the DNA methylation heterogeneity and the overall stemness of cancer cell population. Although the causal explanation between the homogeneity of DNA methylation and the stemness of cancer cells remains unclear, we

can devise an orthogonal method for the computation of the stemness index [15] of a tumor, which has already been shown its prognostic potential in various cancer types. Finally, to facilitate such future studies on the DNA methylation heterogeneity in cancers, we made the DNA methylation heterogeneity profiles of 928 CCLE cell lines computed by Metheor publicly available through Figshare (https://doi.org/10.6084/m9.figshare.21100717.v1).

Metheor is freely available for any users under the GPL-3.0 license. The source code can be anonymously downloaded at the GitHub repository (https://github.com/dohlee/metheor), and the executable is distributed through conda package manager to facilitate the public use of the software (https://anaconda.org/dohlee/metheor). We also provide the codes for the simulation of WGBS reads at the dedicated GitHub repository (https://github.com/jwyang21/simulate_WGBS).

## Supporting information

**S1 Text. Details of the algorithms used in Metheor implementation and simulated data preparation used for benchmark.**
(PDF)

**S1 Fig. Schematic illustration of proportion of discorant reads (PDR).**
(PDF)

**S2 Fig. Schematic illustration of local pairwise methylation discordance (LPMD).**
(PDF)

**S3 Fig. Schematic illustration of methylation haplotype load (MHL).**
(PDF)

**S4 Fig. Schematic illustration of epipolymorphism and methylation entropy.**
(PDF)

**S5 Fig. Benchmarking the running time of Metheor against WSHPackage when only a subset of CpGs are considered.**
(PDF)

**S6 Fig. Robustness of LPMD against the choice of genomic distance window.**
(PDF)

**S7 Fig. Benchmarking the running time and memory usage of Metheor using simulated pseudo-WGBS dataset.**
(PDF)

**S8 Fig. Distribution of the number of CpG pairs at fixed distances in 928 CCLE cell lines.**
(PDF)

**S9 Fig. Association between stemness of cancer cells and other DNA methylation heterogeneity measures.**
(PDF)

**S10 Fig. Promoter PDRs of tumor suppressors and oncogenes.**
(PDF)

## Author Contributions

**Conceptualization:** Dohoon Lee.

**Formal analysis:** Dohoon Lee, Bonil Koo, Jeewon Yang.

**Funding acquisition:** Sun Kim.

**Investigation:** Dohoon Lee, Bonil Koo, Jeewon Yang.

**Methodology:** Dohoon Lee, Bonil Koo, Jeewon Yang.

**Project administration:** Sun Kim.

**Software:** Dohoon Lee, Bonil Koo, Jeewon Yang.

**Supervision:** Dohoon Lee, Sun Kim.

**Validation:** Dohoon Lee, Bonil Koo, Jeewon Yang.

**Visualization:** Dohoon Lee.

**Writing – original draft:** Dohoon Lee, Bonil Koo, Jeewon Yang, Sun Kim.

**Writing – review & editing:** Dohoon Lee, Bonil Koo, Jeewon Yang, Sun Kim.

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
