## [Decision Letter · Decision Letter 0]

4 Dec 2022

Dear Pf. Kim,

Thank you very much for submitting your manuscript "Metheor: Ultrafast DNA methylation heterogeneity calculation from bisulfite read alignments" for consideration at PLOS Computational Biology.

As with all papers reviewed by the journal, your manuscript was reviewed by members of the editorial board and by several independent reviewers. In light of the reviews (below this email), we would like to invite the resubmission of a significantly-revised version that takes into account the reviewers' comments.

We cannot make any decision about publication until we have seen the revised manuscript and your response to the reviewers' comments. Your revised manuscript is also likely to be sent to reviewers for further evaluation.

Sincerely,

Stefan Bekiranov

Guest Editor

PLOS Computational Biology

Lucy Houghton

Staff

PLOS Computational Biology

Reviewer's Responses to Questions

**Comments to the Authors:**

Reviewer #1: Comments are uploaded as an attachment.

Reviewer #2: The manuscript by Lee, et. al developed a fast and lightweight tool for DNA methylation heterogeneity calculation, which made large-scale DNA methylation heterogeneity studies easily feasible for everyone. To illustrate this, the authors have implemented their method on 928 cancer cell lines and revealed potential factors associated with DNA methylation heterogeneity.

Major comments:

1. Cancer cells usually have global low DNA methylation and local high DNA methylation, especially on cancer repressors. Can the authors illustrate the relationships between cancer repressors and DNA methylation heterogeneity?

2. Can the authors get similar results based on the 928 cancer cell lines if they use different DNA methylation heterogeneity measures?

Minor comments:

The authors should make the figures clearer.

**Have the authors made all data and (if applicable) computational code underlying the findings in their manuscript fully available?**

Reviewer #1: Yes

Reviewer #2: Yes

PLOS authors have the option to publish the peer review history of their article (what does this mean?). If published, this will include your full peer review and any attached files.

Reviewer #1: **Yes: **Heng-Chang Chen

Reviewer #2: **Yes: **XIAOLONG CUI
---

## [Decision Letter · Decision Letter 1]

13 Feb 2023

Dear Pf. Kim,

We are pleased to inform you that your manuscript 'Metheor: Ultrafast DNA methylation heterogeneity calculation from bisulfite read alignments' has been provisionally accepted for publication in PLOS Computational Biology.

Best regards,

Stefan Bekiranov

Guest Editor

PLOS Computational Biology

Lucy Houghton

Staff

PLOS Computational Biology

Reviewer's Responses to Questions

**Comments to the Authors:**

Reviewer #1: I have reviewed the resubmission of this revised version of the manuscript. The manuscript has been greatly improved and I appreciate the authors’ efforts to address the concerns that I have raised in the original submission.

This revised version of the manuscript is well-written and clearly explains the background of Metheor, including the theoretical rationale, and algorithms used in this method. Essentially, the authors have strengthened the impact of Metheor on the study of cancer biology and cancer evolution. I do not have any further comments on this manuscript.

Reviewer #2: The authors have addressed all my concerns. The manuscript can be accepted for publication.

**Have the authors made all data and (if applicable) computational code underlying the findings in their manuscript fully available?**

Reviewer #1: Yes

Reviewer #2: Yes

PLOS authors have the option to publish the peer review history of their article (what does this mean?). If published, this will include your full peer review and any attached files.

Reviewer #1: **Yes: **Heng-Chang Chen

Reviewer #2: **Yes: **Xiao-Long Cui

---

## [Editor Report · Acceptance letter]

15 Mar 2023

PCOMPBIOL-D-22-01380R1 

Metheor: Ultrafast DNA methylation heterogeneity calculation from bisulfite read alignments

Dear Dr Kim,

I am pleased to inform you that your manuscript has been formally accepted for publication in PLOS Computational Biology. Your manuscript is now with our production department and you will be notified of the publication date in due course.

With kind regards,

Timea Kemeri-Szekernyes
